# Crystal structures of virus-like photosystem I complexes from the mesophilic cyanobacterium *Synechocystis* PCC 6803

**Yuval Mazor, Daniel Nataf, Hila Toporik, Nathan Nelson***

Department of Biochemistry and Molecular Biology, The George S Wise Faculty of Life Sciences, Tel Aviv University, Tel Aviv, Israel

**Abstract** Oxygenic photosynthesis supports virtually all life forms on earth. Light energy is converted by two photosystems—photosystem I (PSI) and photosystem II (PSII). Globally, nearly 50% of photosynthesis takes place in the Ocean, where single cell cyanobacteria and algae reside together with their viruses. An operon encoding PSI was identified in cyanobacterial marine viruses. We generated a PSI that mimics the salient features of the viral complex, named PSI$^{PsaJF}$. PSI$^{PsaJF}$ is promiscuous for its electron donors and can accept electrons from respiratory cytochromes. We solved the structure of PSI$^{PsaJF}$ and a monomeric PSI, with subunit composition similar to the viral PSI, providing for the first time a detailed description of the reaction center and antenna system from mesophilic cyanobacteria, including red chlorophylls and cofactors of the electron transport chain. Our finding extends the understanding of PSI structure, function and evolution and suggests a unique function for the viral PSI.

## Introduction

Oxygenic photosynthesis, which takes place in cyanobacteria, algae, and plants, provides most of the food and fuel on Earth (*Barber, 2004*; *Nelson, 2011*). Cyanobacterial photosynthetic membranes contain two photosystems, of which PSII mediates the extraction of electrons from water, the initial electron donor, to the plastoquinone pool. PSI mediates electron transfer from cytochrome C553 (or plastocyanin in eukaryotes) to ferredoxin, thereby generating the reducing power needed for $CO_2$ fixation in the form of NADPH. The architecture of oxygenic photosynthesis in cyanobacteria has largely been determined and the structures of both photosystems from thermophilic cyanobacteria have been solved at high resolution (*Jordan et al., 2001*; *Ferreira et al., 2004*; *Nelson and Ben-Shem, 2004*; *Amunts et al., 2007*; *Umena et al., 2011*).

Photosynthetic reaction centers are classified according to their terminal electron acceptor as either type I, with an iron sulfur cluster acceptor, or type II, with a quinone terminal acceptor. PSI, a type I reaction center, presumably evolved from the simpler homodimeric bacterial reaction centers (*Buttner et al., 1992*; *Liebl et al., 1993*; *Baymann et al., 2001*; *Nelson and Ben-Shem, 2005*). We currently know only two versions of the type I reaction centers: the relatively simple bacterial homodimer found in green sulfur bacteria (GSB), *heliobacteria* and *Chloracidobacterium* or the much more complex PSI with its 11–15 subunits found in cyanobacteria and all photosynthetic eukaryotes (*Buttner et al., 1992*; *Hauska et al., 2001*; *Nelson and Yocum, 2006*). Although sequence conservation between the PsaA/B subunits of PSI and the single large subunit of the homodimeric reaction centers is low, their inferred membrane topology is similar, and the sequence conservation around the bound iron sulfur cluster is high, supporting the notion of a common ancestor (*Mulkidjanian and Junge, 1997*; *Blankenship and Hartman, 1998*; *Baymann et al., 2001*; *Mazor et al., 2012*; *Rutherford et al., 2012*).

*For correspondence: nelson@post.tau.ac.il

**Competing interests:** The authors declare that no competing interests exist.

**Reviewing editor**: Werner Kühlbrandt, Max Planck Institute of Biophysics, Germany

**eLife digest** Photosynthesis—the process by which plants and other organisms harness the energy in sunlight—is the source of almost all oxygen, food and fuel on earth. Oxygenic photosynthesis in living cells involves a series of reactions catalyzed by large protein complexes, various other soluble chemicals, and the transfer of electrons from so-called donors to acceptors. The energy in the sunlight is captured by two membrane-embedded protein complexes—photosystem I, which is the most powerful electron donor in nature, and photosystem II—and converted into chemical energy.

Almost half of the world's photosynthesis occurs in the oceans, and is performed by single-celled cyanobacteria and algae. Interestingly, some of the genes that encode photosynthetic enzymes in cyanobacteria are also found in the genomes of viruses that infect these bacteria. It is thought that these viruses can alter photosynthetic pathways in their hosts, but the interactions between these viruses and their hosts are not fully understood.

Now, Mazor et al. have created a photosystem I complex that mimics the viral version of this complex, and have gone on to solve its three-dimensional structure. This mimetic virus-encoded complex was shown to be a 'promiscuous' electron acceptor: this means that, unlike most electron acceptors, it can accept electrons from more than one electron donor.

Further, Mazor et al. solved the structure of photosystem I from *Synechocystis*, a cyanobacterium that lives in fresh water; and found some surprising differences between it and the only other published structure for photosystem I from a cyanobacterium (which was from a species that lives in hot water springs). These included differences in components involved in the electron transfer chain—a series of chemical reactions in which electrons are passed from donor to acceptor molecules—that were thought to be highly conserved. Other differences in the structures allowed Mazor et al. to identify the location of a unique chlorophyll pigment group in the *Synechocystis* photosystem I.

Since *Synechocystis* is commonly used as a model to study photosynthesis, an improved understanding of its photosystem I should lead to further improvements in our knowledge of photosynthesis.

In the earth ocean's, mainly single cell organisms, cyanobacteria or algae, perform oxygenic photosynthesis. One of the most intriguing aspects of marine cyanobacterial photosynthesis is the involvement of various viruses in the process. Hints on this involvement come from the presence of photosynthetic genes in viral genomes (*Mann et al., 2003*; *Sullivan et al., 2006*). Many marine phages encode for either the D1 gene or both D1 and D2 which together make up the reaction center of PSII (*Sullivan et al., 2006*). In addition, several subunits of the NDH1 complex and soluble cytochromes are also found in the viral metagenome (*Philosof et al., 2011*). The presence of these genes suggests that marine viruses have the ability to alter the host photosynthesis pathways in ways, which are still not completely understood. Recently, an operon encoding a complete, albeit minimal, PSI complex was found in the genome of several marine cyanophages (*Sharon et al., 2009*). The viral PSI has a unique gene composition compared to the PSI composition of its hosts. PsaL and PsaI are missing and a unique PsaJ-PsaF (PsaJF) fusion-protein exists.

The absence of both PsaL and PsaI from the viral PSI suggests that it functions as a monomer and does not assemble trimers since both PsaL and PsaI are important for the formation of the cyanobacterial trimer (*Chitnis and Chitnis, 1993*; *Xu et al., 1995*). In eukaryotes, the PsaF subunit is important for electron transfer between plastocyanin (PC) and the PSI complex (*Farah et al., 1995*; *Hippler et al., 1996*). Although the structures of PsaF from cyanobacteria and plants are very similar, the role of PsaF is less well defined in cyanobacteria (*Chitnis et al., 1991*; *Xu et al., 1994a*, *1994b*; *Hippler et al., 1996*). Structurally, the N-terminus of PsaF forms a major 'bump' in the flat, luminal side of PSI. Within this domain, the eukaryotic PsaF contains several additional positively charged residues that stabilize the interaction between PSI and PC (*Farah et al., 1995*; *Hippler et al., 1996*). In the viral PSI the N-terminus of PsaF is deleted and as a result the luminal face of PSI is expected to be relatively flat.

The unique light harvesting and transfer requirements, together with the large number of interconnected subunits, have resulted in a very high degree of structural conservation in PSI complexes (*Ben-Shem et al., 2003*). For example, approximately 90 chlorophyll molecules, bound by the PSI reaction center remained largely unmoved by more than 2 billion years of evolution between the thermophilic

cyanobacterium *Thermosynechococcus elongatus* and the flowering plant *Pisum sativum* (***Jordan et al., 2001***; ***Ben-Shem et al., 2003***; ***Amunts et al., 2007***, ***2010***). A common feature of PSI is the presence of low energy chlorophyll molecules called 'red chlorophylls'. Red chlorophylls are present in the reaction center of cyanobacteria and in the light-harvesting belt of the plant PSI. In cyanobacteria, red chlorophylls can affect the trapping kinetics of excitation energy as shown by measurements done on PSI from various species (***Gobets et al., 2001***). The location of the red chlorophylls within the internal antenna of PSI is still a matter of debate, the inter species variability of the red absorption suggests that they are located in a relatively variable region of PSI. Currently, all the calculations of individual pigment energies are based on the high resolution PSI structure from *Thermosynechococcus elongatus*, which contains a relatively large number of red pigments. New, high resolution structures are needed to compare antenna system across various species.

In this work, we characterized both biochemically and structurally a phage-mimetic PSI^PsaJF constructed in *Synechocystis*. We find that the fused subunit PsaJF is incorporated into the PSI complex and that this incorporation results in a promiscuous PSI, able to accept electrons from respiratory cytochromes. We also present a high resolution model for the *Synechocystis* PSI monomer, which allows us to identify for the first time the changes in the antenna system and electron transport chain (ETC) that exist between *Synechocystis* and *Thermosynechococcus*. The location of red chlorophylls in the thermophilic vs the mesophilic cyanobacteria is discussed.

## Results

### Structure of a trimeric PSI^PsaJF

We hypothesized that the consequences of this viral JF gene product would be a PSI complex with more promiscuous electron acceptor reactions (***Sharon et al., 2009***). Cyanobacteria employ a multitude of electron transfer reactions in their membrane systems. These systems are at least partially insulated by spatial confinement of the various electron donors and acceptors, but the molecular interactions between these donors and acceptors may also contribute to pathway isolation. To explore the promiscuity of viral PSI complexes, we constructed a phage-mimetic PsaJ-PsaF fusion in the easily cultivable cyanobacterium *Synechocystis sp.* PCC 6803. The borders of *PsaJ* and *PsaF* are easily located in the viral gene and the high conservation of the PsaF subunit enabled us to duplicate the salient features of the viral gene into our artificially constructed fusion made from the corresponding *Synechocystis* genes (***Figure 1—figure supplement 1***). *Synechocystis* strains carrying the fused *PsaJF* gene as their sole source of PsaF and PsaJ, grow at wild type rates in heterotrophic conditions and without added carbon. After purifying the PSI complex from the photosynthetic membranes of both wild-type and *PsaJF* strains, we found that the trimer to monomer ratio is similar in both strains, indicating that the assembly and stability of PSI^PsaJF is not disturbed (***Figure 1A***). The subunit composition of the purified complex showed that it indeed contained the fused PsaJF subunit in lieu of PsaJ and PsaF (***Figure 1B***).

To get a more detailed view of the PsaJF subunit in PSI^PsaJF, we crystallized PSI^PsaJF and obtained diffraction data to 3.8 Å resolution (***Table 1***). The crystal structure of the PSI^PsaJF complex was solved by molecular replacement, first by using the previously determined cyanobacterial structure as a search model (PDB ID 1JBO) and later with a higher resolution model from *Synechocystis* (see 'Results' below). The model reveals three monomers in the asymmetric unit. A complete trimeric complex is formed from monomers positioned in three unit cells (***Figure 1C***). As seen in ***Figure 2A,B***, the overall architecture of PSI from *Synechocystis* is highly similar to the *Thermosynechococcus elongatus* PSI with two noticeable exceptions, the fused PsaJF subunit and PsaX, which as expected, is missing (the genome of *Synechocystis* PCC6803 does not contain a PsaX homolog).

Our trimeric PSI structure is the first look at a mesophilic cyanobacterial PSI. As such, even at this relatively low resolution, some interesting features are apparent. The structure clearly shows the presence of the fused subunit integrated in the PSI^PsaJF complex (***Figure 2B***). Deleting the N-terminus of PsaF resulted in a flattened plateau on the luminal side of the complex (***Figure 2B,C***). Three chlorophyll molecules and four carotenoids are coordinated by PsaJ and PsaF in the *Thermosynechococcus* PSI structure (***Figure 2C***). We find that all of these ligands are probably lost during the assembly of PSI^PsaJF and could not be traced in our model (***Figure 2C***). In spite of the relatively low resolution, we find that the overall organization of the other cofactors in PSI is hardly affected in PSI^PsaJF demonstrating the mild effect of the JF mutation on the overall organization and integrity of the complex.

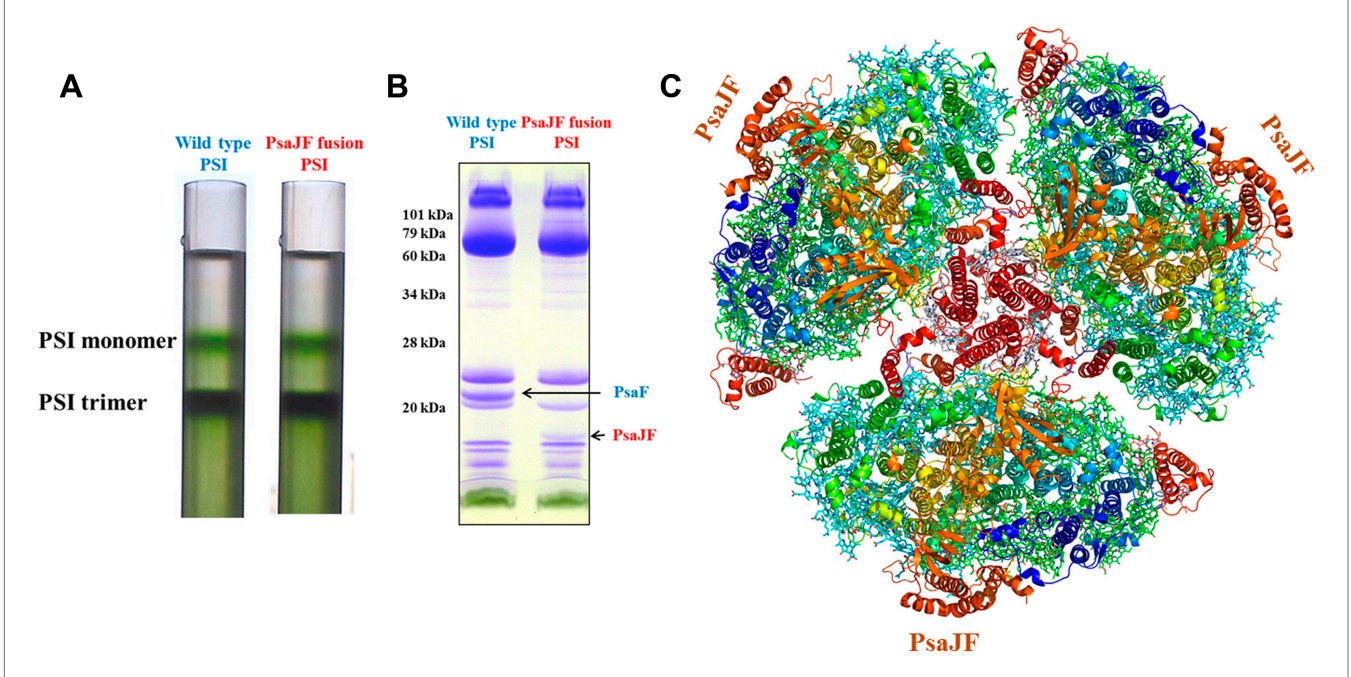

**Figure 1**. The crystal structure of a PsaJF fusion containing PSI complex. (**A**) Sucrose gradients of chlorophyll containing complexes from either wild type or PsaJF containing bacteria showing similar trimer to monomer ratios in both strains. (**B**) An SDS-PAGE gel showing the polypeptide composition of the trimeric PSI fractions from the sucrose gradient showing the presence of the PsaJF fusion (which was confirmed by MS analysis) in the purified PSI complex. (**C**) The PSI trimer structure viewed from the stromal side of the membrane showing the fused subunit assembled in the complete complex.

The following figure supplements are available for figure 1:

**Figure supplement 1**. Multiple sequence alignment showing the domain architecture of the viral JF fusion sequences compared to the cyanobacterial PsaF and PsaJ sequences.

## PSI$^{PsaJF}$ is a promiscuous electron acceptor

Next we tested whether PSI$^{PsaJF}$ is a promiscuous electron acceptor. We measured the half-life of PSI reduction by a mammalian respiratory cytochrome c. As seen in **Figure 3A**, P700 in PSI$^{PsaJF}$ is reduced much faster by a respiratory cytochrome c than in the wild-type complex. However, PSI$^{PsaJF}$ is reduced by its native electron donor cytochrome c553 (CytC553) with similar kinetics compared to the wild-type PSI (**Figure 3B**). This is in contrast to eukaryotic PSI where the N-terminus of PsaF was shown to be important for the electron donation reaction between plastocyanin and PSI (**Hippler et al., 1996**). The similar electron donation rates observed with CytC553 are in agreement with previously published results obtained from PsaF and PsaJ deleted strains (**Xu et al., 1994a**, **1994b**). When the light-dependent oxidation of cytochrome c was followed we obtained similar results. Compared to the wild-type complex, PSI$^{PsaJF}$ had faster cytochrome oxidation kinetics when respiratory cytochrome was used (**Figure 3C**), and similar kinetics when native CytC553 was used (**Figure 3D**). The similarity between the reaction kinetics of PSI reduction together with the similar cellular growth rates of wild-type and PSI$^{PsaJF}$ strains shows that deleting the N-terminus of PsaF causes no gross perturbation to the interaction between PSI and CytC553, that is, the N-terminus of PsaF does not play any positive role in this interaction in *Synechocystis*. The N-terminus of PsaF can act as a negative regulator of some electron donation reactions, especially from positively charged donors such as respiratory cytochromes. We suggest that this was the original function of PsaF in a more primitive form of PSI and only later, as PSI evolved, the binding site was refined to its final configuration. In the viral PSI, such promiscuity can be beneficial when the viral reaction center encounters the highly variable cytochromes found in marine cyanobacteria (**Mazor et al., 2012**) or alternatively to allow PSI to accept electrons from respiratory activities when PSII activity is compromised.

**Table 1.** X-ray data collection and refinement statistics

| Data collection | PsaJF trimer | PSI monomer | |
|---|---|---|---|
| Beamline | ESRF–ID29 | ESRF–ID29 | SLS–PXI–X06SA |
| Wavelength (Å) | 0.97625 | 0.97625 | 1 |
| Resolution (Å) | 30–3.8 | 30–2.8 | 30–3 |
| Measured reflections | 426,209 (63,882) | 419,672 (61,558) | 396,647 (57,225) |
| Unique reflections | 113,221 (16,536) | 91,895 (13,209) | 72,095 (10,585) |
| Rpim (%) | 6.8 (71) | 7.5 (124) | 5.6 (77.3) |
| $<I>/<\sigma(I)>$ | 9.9 (1.2) | 8 (1.3) | 10 (1.2) |
| Completeness | 98.9 (99.5) | 99.4 (99.1) | 96.8 (98.3) |
| Redundancy | 3.8 (3.9) | 4.6 (4.7) | 5.1 (5.4) |
| Space group | P 21 | P 21 21 21 | P 21 21 21 |
| Unit cell dimensions | | | |
| a, b, c (Å) | 214, 134, 220 | 120, 173, 179 | 120, 174, 179 |
| α, β, γ (°) | 90, 111.1, 90 | 90, 90, 90 | 90, 90, 90 |
| Refinement statistics | | | |
| Resolution (Å) | 30–3.8 | 30–2.8 | 30–3 |
| Rwork/Rfree | 25.9/29.7 | 21/24 | 24.4/28 |
| No. of chains | 30 | 9 | 9 |
| No. of ligands | 360 | 119 | 116 |
| Average B-factor (Å²) | 128 | 85 | 90 |
| R.M.S deviations | | | |
| Bond Angles | 1.7 | 1.7 | 1.7 |
| Bond lengths | 0.014 | 0.005 | 0.004 |
| Ramachandran statistics | | | |
| Favored region % | 93.8 | 93.8 | 93.2 |
| Allowed region % | 5.1 | 5.9 | 6.3 |
| Outlier region % | 1.1 | 0.3 | 0.5 |
| clashscore | 5.5 | 3.8 | 4.6 |

## High-resolution structure of a PSI monomer

The small subunits PsaF, J, K, L, I, and M of the PSI complex can each be deleted without observing any severe growth defects in cells (*Chitnis and Chitnis, 1993*; *Xu et al., 1995*; *Naithani et al., 2000*). Their conservation in all cyanobacterial species, arguably the most diverse group of microorganisms, suggests that over evolutionary time periods losing them is detrimental. Different small subunits, however, are functionally connected. For example, in cells lacking PsaJ, PsaF is only partially incorporated into the complex (*Xu et al., 1994a*). A similar situation is seen in PsaI deleted cells, where PsaL is only partially assembled into the complex (*Xu et al., 1995*). In both the cases, a 'simple' single transmembrane (TM) helix (PsaJ or PsaI) seems to be necessary for the assembly of a more complex subunit (PsaF or PsaL, respectively).

To improve the diffraction from our crystals, and increase the observation to parameter ratio of our model, we purified and crystallized the monomeric form of PSI. This approach was attempted previously, but did not bear fruit (*Jekow et al., 1995*, *1996*; *Fromme and Mathis, 2004*). We inserted 10 histidine residues at the C-terminus of PsaL, similar to an approach previously shown to completely disrupt trimer formation without adversely affecting PSI activity (*Tang and Chitnis, 2000*). As expected, extending the C-terminus of PsaL caused no significant effect on cell growth or PSI activity and resulted in a homogenous population of monomeric complexes that were readily purified and crystallized (*Figure 4—figure supplement 1A,B*).

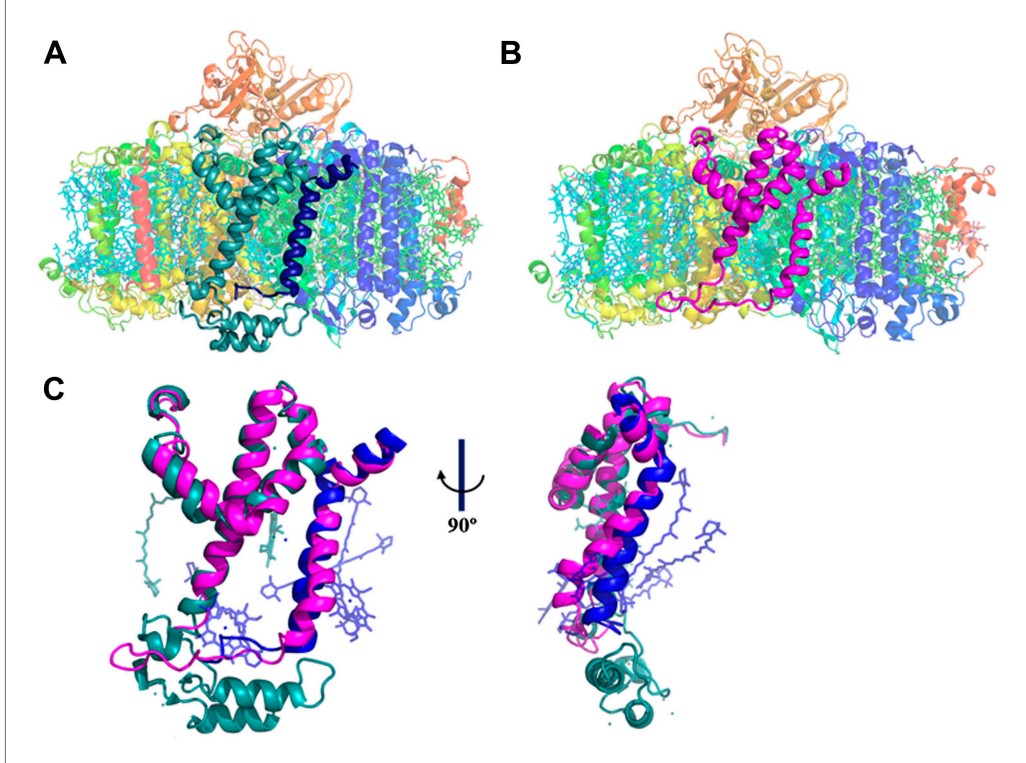

**Figure 2**. A JF containing PSI complex has a luminal plateau. (**A**) PSI from *Thermosynechococcus* with PsaJ (Blue) and PsaF (cyan) shown from the membrane plane, the N-terminus of PsaF extends as two alpha helices into the luminal space. PsaX is colored red. (**B**) The PsaJF (magenta) containing PSI complex from *Synechocystis* showing the flatter luminal plateau created from the removal of the N-terminus of PsaF. (**C**) Superposition of the JF subunit from *Synechocystis* (magenta) on the PsaF (cyan) and PsaJ (blue) subunits from *Thermosynechococcus*, the overall structure of the fused subunit remains undisturbed.

Fortuitously, PsaL and PsaI were missing from the final purified complex (*Figure 4—figure supplement 1C*), making the subunit composition of the crystallized PSI very similar to the viral PSI complex (the only difference is the single TM subunit PsaM which is found in our preparation and is absent from the viral PSI operon). This shows that the subunit composition of the phage complex is sufficient to produce a stable and active PSI complex.

The crystals obtained from this monomeric form of PSI belong to a higher symmetry space group (P212121), had a reduced cell volume (~$3.75 \times 10^6$ Å³) and, most importantly, diffracted to a resolution of 2.8 Å. At this resolution, the entire complement of ligands and cofactors is revealed, and it is feasible to carry out a detailed comparison between the thermophilic and mesophilic complexes.

## Electron transport chain

The available structures of PSI from thermophilic cyanobacteria and higher plants show complete conservation in all the components of the electron transport chain (ETC) from P700 to the iron sulfur clusters (*Jordan et al., 2001*; *Ben-Shem et al., 2003*; *Amunts et al., 2007*). In light of this complete conservation, we were surprised to discover a large rearrangement of the isoprenoid tail of the phylloquinone ($Q1_A$) coordinated by PsaA (*Figure 4B*, *Figure 4—figure supplement 2*). This movement is accompanied by rearrangement of the phytol tails of chlorophylls A06 and A40 (ligands are named according to Jordan, et al. whenever possible [*Jordan et al., 2001*]). In spite of these large movements, the positions of the quinone heads and chlorine rings are highly conserved between all known PSI structures (*Figures 4B and 5A*). The movements in $Q1_A$ and the surrounding chlorophylls appear to originate from two amino acid changes that occurred in the PsaJ subunit between *Synechocystis* and *Thermosynechococcus*. Ala J16 in *Thermosynechococcus* changed to Met J16 in *Synechocystis*, making contacts with terminal carbon atoms of $Q1_A$ (*Figure 4C*). Similar contacts are seen in the

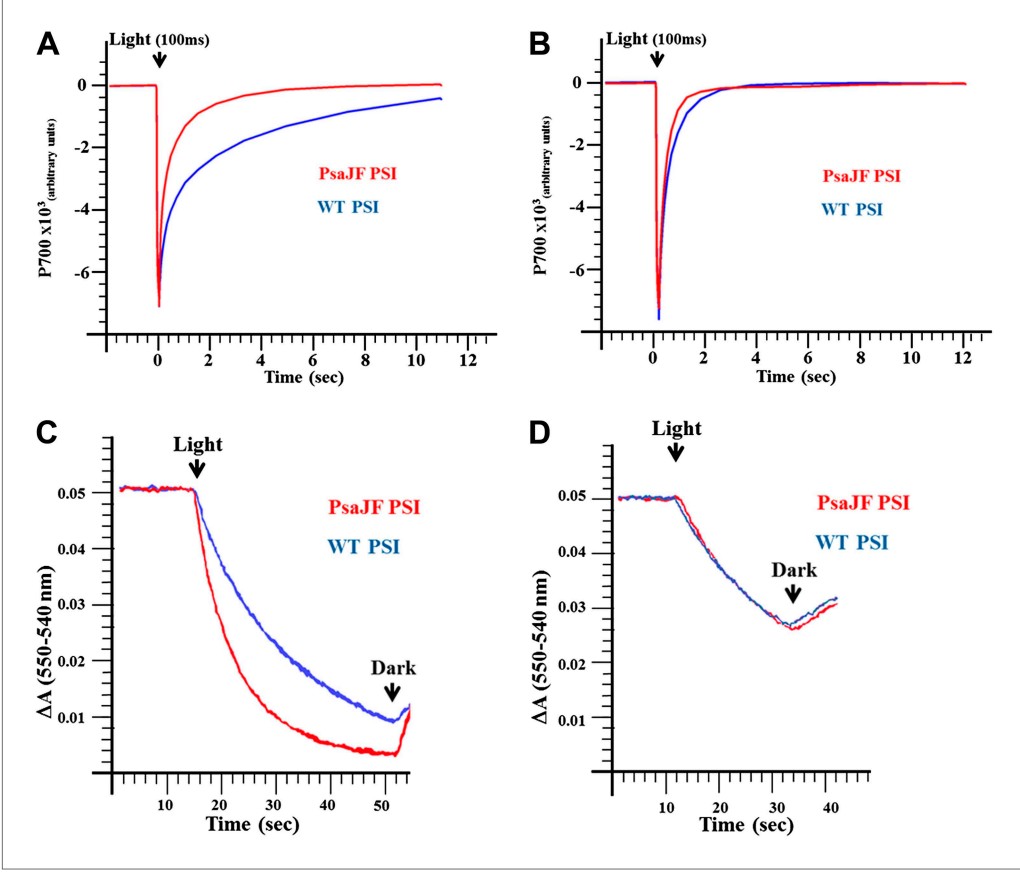

**Figure 3**. PSI$^{PsaJF}$ is a promiscuous electron acceptor. (**A**) P700 reduction kinetics (after a 100 ms pulse of orange light) showing faster reduction kinetics of a PSI$^{PsaJF}$ complex with respiratory CytC (red line) compared to the wild-type complex (blue line). (**B**) Native cytochrome C553 reduces both wild-type and PSI$^{PsaJF}$ complexes at similar rates. (**C**) Plots of the cytochrome oxidation (ΔA 550 nm–540 nm) under continuous white illumination by PSI, showing faster oxidation of a respiratory cytochrome by PSI$^{PsaJF}$ (red trace) compared to the wild-type PSI (blue trace). (**D**) Native CytC553 is oxidized with similar kinetics by both complexes.

*Thermosynechococcus* Q1$_A$ terminal carbon atoms, but these contacts are made with Met 19 in PsaJ, which changed to Leu 19 in *Synechocystis* (*Figure 4C*). Interestingly, this interaction between the terminal carbon atoms of the isoprenoid tail and methionine is present in a number of cyanobacterial species (*Figure 4—figure supplement 4*). It is interesting to note that most of the other contacts stabilizing the tail conformation of Q1$_A$ mainly involve interactions between the hydrophobic tails of other ligands.

Another noticeable feature of the ETC is the conservation of water molecules position around Fx (*Figure 4—figure supplement 3*). This water pocket is positioned between Q1$_B$ and Fx and can potentially modify the ET rates between the A and B branches of PSI. Mutations in the coordinating tryptophan residue greatly affect the ET rates between Q1$_B$ and Fx (*Ali et al., 2006*).

Changes in the orientation of the isoprenoid tail can fine-tune the ETC. The fact that small, less conserved, subunits, such as PsaJ, are involved in changing these orientations, provides relaxed evolutionary constraints that facilitate faster exploration of the available changes (*Peisajovich et al., 2006*). To summarize, the components of the ETC are extremely conserved over all known PSI structures. However, even small changes in the orientation of components can play a part in fine tuning ET processes. Most of the studies on the ETC are performed in *Synechocystis* (*Savitsky et al., 2013*) and assume that all of the factors involved occupy identical positions as shown in *Thermosynechococcus* or higher plants, this may lead to somewhat erroneous conclusions. The surprising observation of a conserved water cluster positioned along the ET path between Q1B and Fx suggests that this feature has a functional role.

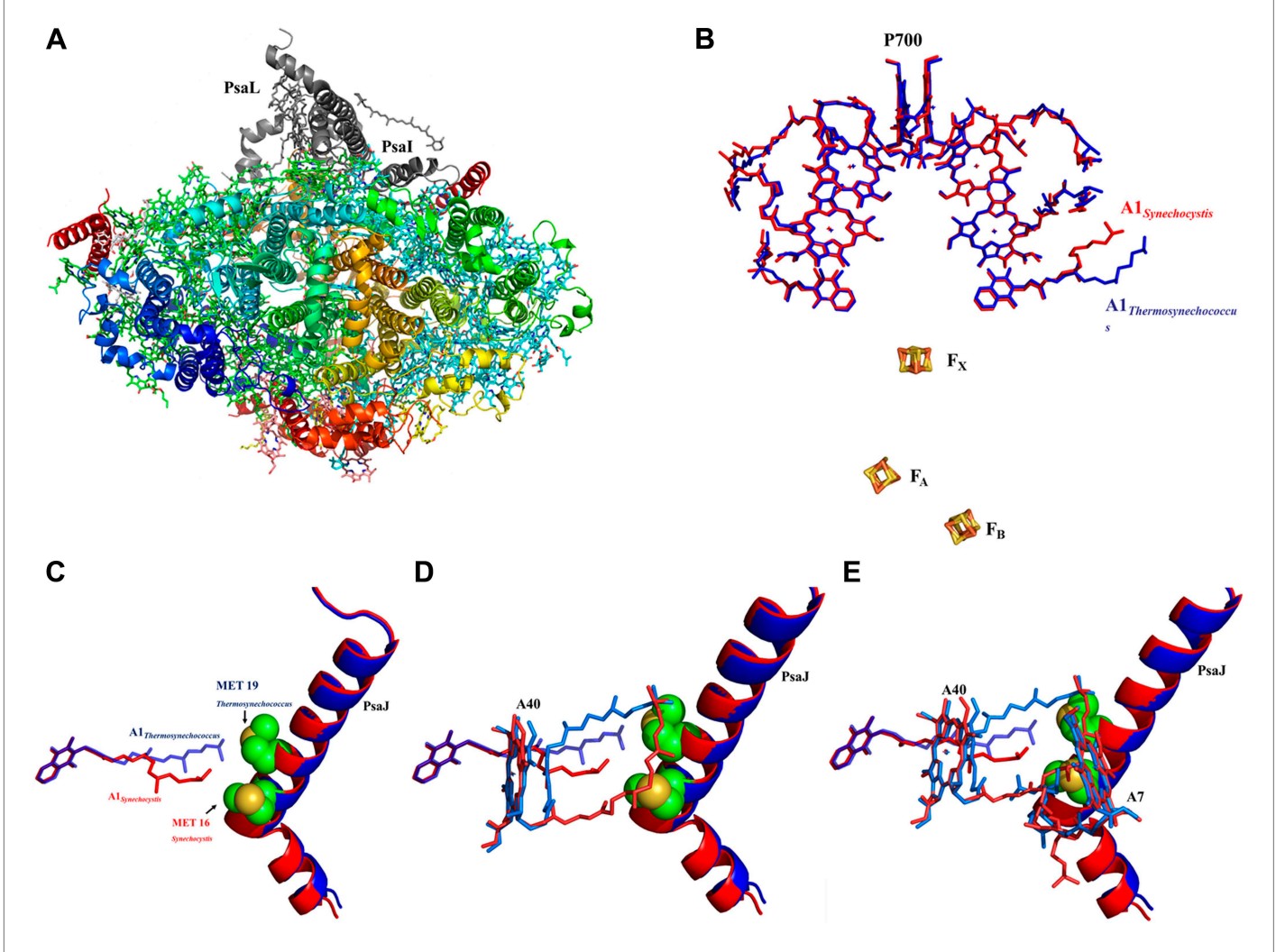

**Figure 4**. High resolution structure of a monomeric PSI complex from *Synechocystis*. (**A**) A luminal view of the monomeric PSI complex. PsaL and PsaI that are missing from our crystal are shown in gray. (**B**) Comparison of the ETC of *Synechocystis* (red) to that of *Thermosynechococcus* (blue) reveals near perfect superposition in all the components with the clear exception of the quinone A1 isoprenoid tail. (**C**) Superposed PsaJ from *Synechocystis* (red) and *Thermosynechococcus* (blue) showing the coordinating methionine residue (carbons in green and the sulfur atom in yellow). (**D**) The phytol tail of chlorophyll A40 is reoriented in *Synechocystis* in order to accommodate the movement of the quinone tail. (**E**) The phytol tail of Chlorophyll A1 in *Synechocystis* shifted to accommodate the PsaJ MET16 change.

The following figure supplements are available for figure 4:

**Figure supplement 1**. Purification of a monomeric PSI from a *Synechocystis psaI* mutant.

**Figure supplement 2**. Electron density map of Q1$_A$ showing the different position of the A1 quinone from *Synechocystis* (in red).

**Figure supplement 3**. Conservation of a water pocket lining the electron path between Q1$_B$ and Fx.

**Figure supplement 4**. Multiple sequence alignment (MSA) of PsaJ sequences from various cyanobacteria.

## Light-harvesting and red chlorophylls

The polypeptide chains of PSI coordinate more than a 100 pigments, which funnel excitation energy to P700 with near absolute quantum efficiency (*Croce and van Amerongen, 2013*). A common theme in PSI is the occurrence of red chlorophylls, which absorb (and fluoresce) light at longer wavelengths

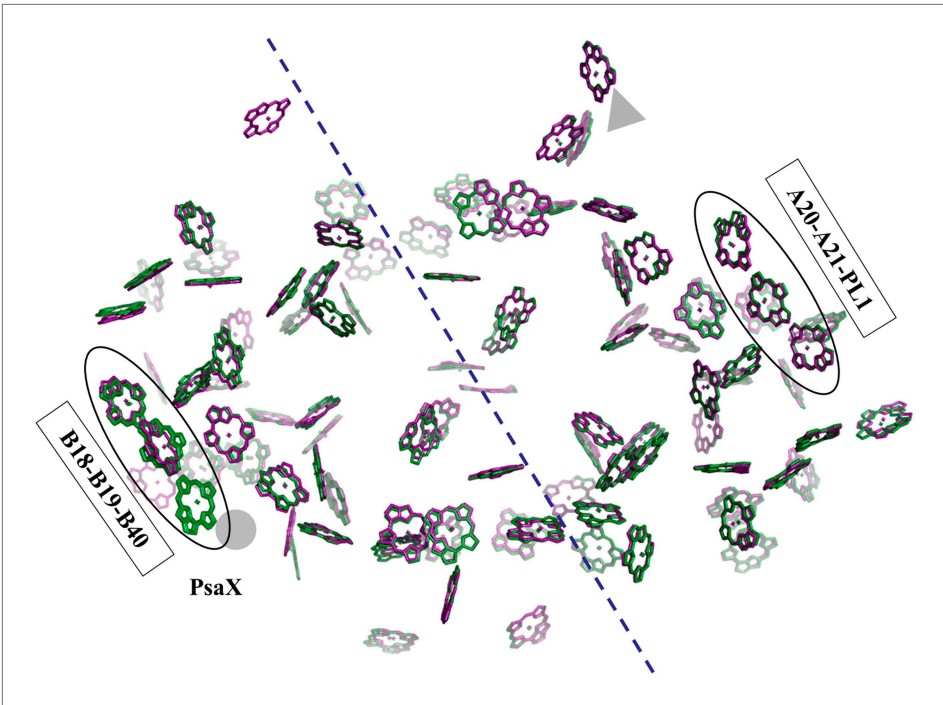

**Figure 5**. Pseudo C2 symmetry in the antenna system of *Synechocystis*. A depth cued image comparing the overall organization of the PSI antenna from *Synechocystis* (green) or *Thermosynechococcus* (purple) viewed from the stromal side of the membrane, only chlorine rings are shown for clarity. The C2 pseudosymmetry axis is shown in blue. PsaX is shown as gray filled circle. The new chlorophyll detected in *Synechocystis* (B40) is clearly seen next to PsaX which precludes its binding in *Thermosynechococcus*.

The following figure supplements are available for figure 5:

**Figure supplement 1**. Red chlorophylls conformations in *Synechocystis* monomeric and trimer PSI.

---

than P700 (*Palsson et al., 1998*; *Gobets et al., 2001*). Red chlorophylls slow the trapping kinetics of P700, however, the quantum yield of the complex is not affected by this at room temperature; at higher light intensities red traps may have a photoprotective role. In addition, red chlorophylls increase the absorbance cross section of the entire complex, especially in shaded environments.

There is considerable variability in the strength of the red absorption between various cyanobacterial species and in particular *Synechocystis* is known to have very low levels of red chlorophylls in comparison to *Thermosynechococcus* (*Gobets et al., 2001*). The exact identity of the red pigments within the reaction center of PSI is unknown and the clear difference in their content between *Synechocystis* and *Thermosynechococcus* presents us with an opportunity to identify these chlorophylls. Based on ring-to-ring distances and dipole orientation *Jordan et al. 2001* identified one chlorophyll trimer (B31-B32-B33) and three dimers (A32-B7, A38-A37, B37-B38) as candidates for strongly coupled pigments in the reaction center. We find that the ring location and the local environment of dimer A38-A39 remained virtually unchanged between *Synechocystis* and *Thermosynechococcus* (*Figure 5—figure supplement 1A,B*). The chlorine rings of dimer B37-B38 occupy very similar positions in both photosystems including the coordinating water molecule of B37 that can clearly be seen in our maps (*Figure 5—figure supplement 1C*). The phytol tails of both chlorophylls shifted considerably, however because of this dimer's proximity to PsaL definite conclusions are not possible (*Figure 5—figure supplement 1C,D*).

The ring positions of the A32-B7 dimer shifted slightly from *Thermosynechococcus* to *Synechocystis*, and the side chain coordinating the magnesium of B7 changed from glutamine to histidine. These changes can contribute greatly to the $Q_y$ position of the pigment (*Wientjes et al., 2012*). However, this dimer is also located close to PsaI and PsaL, and therefore may have been affected by their exclusion from the complex (*Figure 5—figure supplement 1E,F*). Indeed, in the PSI$^{PsaJF}$ model, the positions of both rings are slightly shifted toward the position observed in *Thermosynechococcus*. This dimer

may be sensitive to the oligomerization state of the complex, as it has been shown that some red absorption is lost upon monomerization (*Gobets et al., 2001*), a phenomenon which is common to both *Synechocystis* and *Thermosynechococcus*. A monomeric structure that includes both PsaI and PsaL will resolve this question.

In contrast, it is very clear that chlorophyll B33 is completely missing from the stacked trimer observed in *Thermosynechococcus* (*Figures 5 and 6*). This is accompanied by a small deletion in the PsaB loop, which supports B33. A similar deletion is found in approximately third of the cyanobacterial PsaB sequences, as well as the PsaB sequences in eukaryotes (*Figure 6—figure supplement 1*), where the corresponding chlorophyll has shifted substantially (*Ben-Shem et al., 2003*; *Amunts et al., 2007*). The ring distances and orientation between B31 and B32 hardly changed from *Thermosynechococcus* to *Synechocystis*. These findings strongly support the B31-B32-B33 trimer as the strong red absorber in *Thermosynechococcus* and suggest that the one of remaining dimers, either B37-B38 or B31-B32, is responsible for the residual red absorbance seen in *Synechocystis*.

Photosystem I contain another chlorophyll trimer that is coordinated by PsaA (A20-A21-PL1). The ring-to-ring distances between the chlorophylls of this trimer do not suggest direct coupling, but, surprisingly, our model revealed the presence of an additional chlorophyll trimer, related by the pseudo C2 symmetry axis of the reaction center (*Figures 5 and 6A,B*). The new trimer is formed by an additional chlorophyll, B40, which is coordinated by the head group of a phospholipid (one of three present in PSI) (*Figure 6A,B*). The phosphate oxygen atom lays 2.8 Å from the magnesium atom of B40. This coordination of chlorophylls by lipids is reminiscent of the situation in light-harvesting

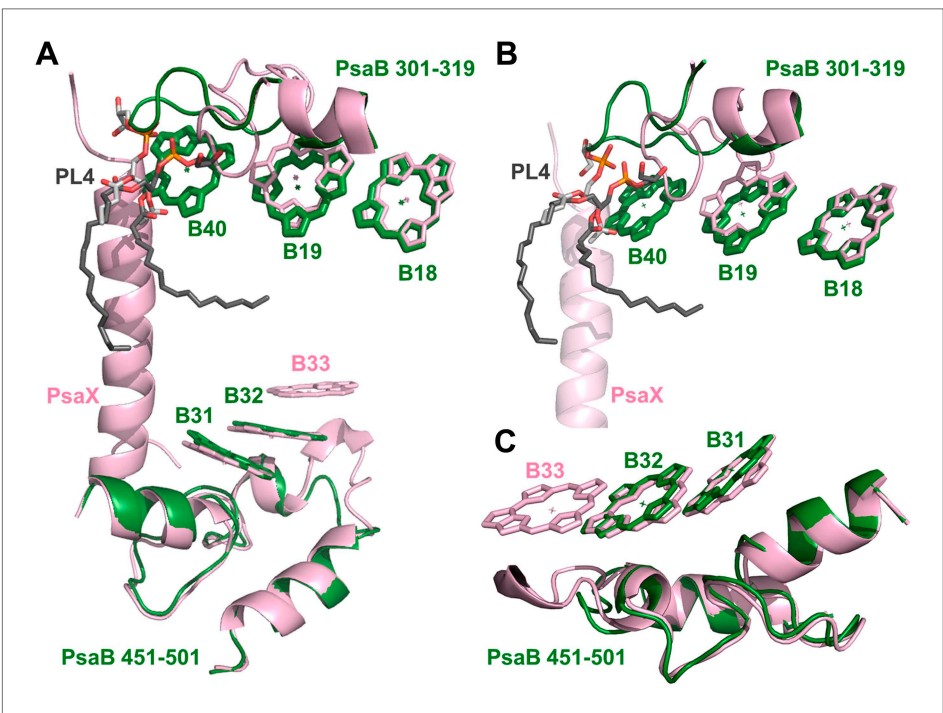

**Figure 6**. Complementing structural rearrangements on opposite sides of PSI in *Synechocystis*. (**A**) The luminal-side chlorophyll trimer (B31-B32-B33) is missing in *Synechocystis*, while chlorophyll B40 forms a stromal chlorophyll trimer (*Synechocystis* in green, *Thermosynechococcus* in light pink). The PsaB loop on the stromal side (PsaB 301-319) extends to coordinate chlorophyll B40 and A phospholipid (shown in dark gray), while the luminal side PsaB loop (PsaB 451-501) is shortened in *Synechocystis* and chlorophyll B33 is lost as a result. (**B**) A close up on the stromal side changes showing the large conformational change in PsaB. In spite of this the phospholipid is clearly observed. (**C**) Loss of the B31-B33-B33 chlorophyll trimer in *Synechocystis*, the shortened PsaB loop together with the remaining chlorophylls in *Synechocystis* is shown.

The following figure supplements are available for figure 6:

**Figure supplement 1**. High variability in the B31-B32-B33 supporting loop of PsaB.

complex II (LHC-II) (*Standfuss et al., 2005*). In contrast, CP29, which is highly homologous to LHC-II, does not contain such a configuration and this probably contributes to the different non photochemical quenching properties of both proteins (*Pan et al., 2011*).

The position of this new chlorophyll (B40) is occupied by subunit PsaX in *Thermosynechococcus* (*Figure 6*). Surprisingly, and in spite of the large structural changes observed in this region, this lipid is still clearly visible, this high degree of conservation demonstrates the important role of the PSI lipids. The fact that the newly discovered chlorophyll trimer in *Synechocystis* relates to its counterpart by pseudo C2 symmetry (*Figure 6*) suggests that this configuration preceded the association of PsaX, which may have been an adaptation to high temperature. Possibly the most important role of PsaX is to protect this lipid at higher temperatures. The occurrence of this chlorophyll trimer suggests that these trimers have a structural role and the appearance of the red absorption results from fine-tuning of their position. It is likely that the pigment environment contributes significantly to its spectral properties.

## Discussion

Photosystem I is one of the most complex enzymes in nature. Even though the evolution of photosynthesis must have taken a complex path, today we can only see a limited snapshot of this path. The high degree of sequence conservation and the importance of pigments to the photosynthetic process further limit our ability to deduce function from mere sequences.

### Viral reaction centers

The prevalence of photosynthetic genes in marine cyanophage genomes suggests that photosynthesis and respiration play an important part in the phage–host interaction. Aside from the PSI operon, genes for other subunits of the major complexes of the photosynthetic electron transport chain reside in phages including PSII, NDH-I, F-ATPase (*Philosof et al., 2011*). Small soluble proteins like cytochromes and ferredoxins are also abundant (*Sharon et al., 2011*). A common property of these genes is that they cluster together as a distinct group when compared to their cyanobacterial counterparts (*Zeidner et al., 2005*). In some cases, it is possible to suggest that viral subunits modify the flow of electrons in the photosynthetic chain based on the known function of the cyanobacterial counterpart. This is the case with the virally encoded NDH-I subunits, where the function of the native, bacterial complex in cyclic electron flow is established.

In this study, we show that a phage specific mutation, removal of the N-terminus of PsaF, results in a promiscuous PSI complex. It is known that the properties of the small electron donor (either PC or CytC553) largely determine its mode of interaction with PSI (*Hervas et al., 2005*). Marine cyanobacteria contain an especially diverse set of soluble cytochromes with variable physical properties (*Mazor et al., 2012*). How this diversity is manifested with respect to cellular respiration and photosynthesis is a matter for future experiments. However, our findings suggest that photosynthesis performed during phage infection and catalyzed (at least partially) by phage proteins ('Viral photosynthesis') utilize very different cytochromes efficiently and will not be sensitive to the nature of the electron donor. The existence of both NDH-I subunits and a promiscuous PSI operon in a single phage genome suggests that cyclic electron flow is enhanced during phage replication (*Philosof et al., 2011*). This can be achieved by increasing the flow of electrons into the photosynthetic electron transport chain through the NDH–I complex or by utilizing electrons from respiratory cytochromes or both. It is important to consider the two most common phage photosynthetic proteins, D1 and D2, in that light. The suggested function of virally encoded D1 and D2 is to replenish the rapidly damaged host D1 and D2 proteins during infection (*Mann et al., 2003*; *Alperovitch-Lavy et al., 2011*). This seems to partially contradict the notion of increased cyclic electron flow in infected cells but it is probable that some basal level of linear electron flow in the cells plays a major part in phage multiplication. Detailed studies performed in phage-infected cells are needed to uncover the facts needed to support these hypotheses.

We suggest that the phage-encoded PSI complex with its unique gene composition adds an important step to the scenario of PSI evolution from a primordial homodimeric entity to the current highly evolved complexes (*Nelson, 2011*; *Mazor et al., 2012*). In this scenario, the formation of the primordial heterodimer allowed for the evolution of the unique properties of the peripheral subunits, trimer formation in the case of PsaL and interaction with electron donors in the case of PsaF. The specificity and complexity of biological machines increases along their evolutionary path and PSI is a prime example of this property. Cyanobacterial PSI is much more complicated and intricate than its ancestral homodimer and the plant PSI adds additional levels of complexity over the cyanobacterial complex. The preservation of simpler forms of

PSI is therefore quite unique and stems from the relatively simple life cycle of phages and from the unique evolutionary requirements associated with this life cycle. For example, some marine phages from the *Myoviridae* family can infect different species of cyanobacteria (*Sullivan et al., 2003*) and this creates unique requirements on their genes such as utilizing the different chlorophyll species found in *Prochlorococcus* and *Synechococcus*. The viral complex preserves some of the properties of the common PSI ancestor, lost from both the cyanobacterial and plant PSI. One of these properties appears to be promiscuity for electron donors as exemplified by the fused PsaJF subunit. The incredible richness uncovered by ocean genomic surveys suggests that further sampling will reveal further examples of ancestral forms of PSI and other unique versions of highly conserved genes.

### Mesophilic cyanobacterial PSI

The two structures described in this work provide the first detailed look at the *Synechocystis* PSI complex. The two most prominent differences between *Synechocystis* and *Thermosynechococcus* are the longer trapping time for excitation energy and the stronger red absorbance found in *Thermosynechococcus*. Both of these differences are probably related to the number of red-shifted pigments found in PSI. This study provides definite proof for the absence of the putative 'red' chlorophyll trimer in *Synechocystis* and narrowed the number of coupled pigment dimers to two. Further calculations should help identifying other, more subtle alteration in pigment energy along the antenna system. Structurally, the *Synechocystis* PSI complex revealed the PsaX-less conformation of the antenna and it is evident that this configuration is closer to the plant PSI inner antenna and thus can be considered the ancestral arrangement.

One of the most surprising findings is the conserved lipid binding by the PsaB loop close to PsaX (*Figure 6*). This is the region of PSI that underwent the most drastic changes between *Synechocystis* and *Thermosynechococcus*, however the presence of the lipid is not affected by them. This suggests that the function of this lipid (and by inference the function of the other PSI lipids) is related to assembly process of PSI, possibly to support reinsertion into the membrane after the synthesis of an extended stromal loop. The absence of any sequence identifiable binding site for this lipid stems from the hydrophobic interactions made between its hydrocarbon tails and several chlorophyll molecules without involving any protein atoms. This highlights the significance of detailed structural information in elucidating the biology of complex biological assemblies.

A cluster of water molecules located between Fx and $Q1_B$, conserved between *Thermosynechococcus* and *Synechocystis* was revealed. The two branches of the PSI ET chain are not equivalent, the PsaB located branch being faster. This water cluster may modulate the contribution by Tryptophan PsaB 664 to the electron transfer between $Q1_B$ and Fx. The high structural conservation observed in this region implies that the final answers can only be reached using ever more detailed models and refined calculation methods which will include all participating atoms.

## Materials and methods

### Culture conditions

Cyanobacteria were cultured in BG11 medium supplemented with 10 µg/ml Ferric ammonium citrate, 5 mM glucose and 10 µg/ml chloramphenicol under continuous white light (~40 µE) in 30°C.

### Strain construction

*Synechocystis PsaJ* (amino acids 1–40) and *PsaFΔN* (amino acids 84–165) were amplified by PCR and fused together with a Cys-Ser-Cys linker between them. The *PsaJF* gene was fused to the native *PsaF* promoter together with 300 bp of upstream and downstream sequences to direct its recombination to the genome of *Synechocystis*. The Kanamycin resistance gene was cloned just downstream to the *PsaJF* gene. Primer sequences are available in *Supplementary file 1*.

Transformation of *Synechocystis* sp. PCC6803 was performed according to standard protocols. Gene replacement constructs were generated in pGEM or pJET, and used directly for transformation. All transformants were streaked on increased antibiotic concentrations for a minimum of three times, and verified by PCR.

### Photosystem I purification

20–40 l of cells were grown in regular BG11 (supplemented with 10 µg/ml ferric ammonium citrate and 5 mM glucose, plus 10 µg/ml chloramphenicol for strains with His-tagged PsaL) under light intensity of ~40 µE at 30°C. Cells were harvested using centrifugation, and washed once by STN1 buffer (30 mM

Tricine-NaOH pH 8, 15 mM NaCl, 0.4 M sucrose). Finally, cells were resuspended in 50 ml of STN1, and broken by an Avestin EmulsiFlex-C3 (three cycles at 1500 psi). The lysate was cleared by centrifugation in a SS34 rotor for 10 min at 12,000 rpm. Membranes in the supernatant were pelleted using ultracentrifugation (Ti70 rotor, 45,000 rpm for 2 hr), and resuspended in 50 ml STN2 (30 mM Tricine-NaOH pH-8, 100 mM NaCl, 0.4 M sucrose). After resuspension in STN2, the membranes were incubated on ice for 30 min, then collected again (Ti70 rotor, 45,000 rpm, 2 hr), and resuspended in approximately 15 ml of STN1. n-Dodecyl β-D-maltoside (DDM, Affymetrix, http://www.affymetrix.com/estore/index.jsp) was added to the membranes (from a 10% stock solution in water) at a 15:1 DDM to chlorophyll ratio. The suspension was gently mixed by hand a few times then incubated on ice for 30 min. After solubilization, the insoluble material was discarded using ultracentrifugation (Ti70, 45,000 rpm, 30 min). The solubilized membranes were loaded onto a DEAE column (DE52; Whatman http://www.gelifesciences.com/webapp/wcs/stores/servlet/catalog/en/GELifeSciences/brands/whatman/), column volume was adjusted to the chlorophyll content of the sample (1.5 or 1.2 ml DEAE per 1 mg chlorophyll for trimer or monomer preparations, respectively). The column was eluted using a linear NaCl gradient (15–200 mM NaCl for monomer preparation; 15–350 mM NaCl for trimer preparations) in 30 mM Tricine-NaOH pH 8, 0.2% DDM. Dark green fractions were collected and precipitated using PEG3350 (Hampton research http://hamptonresearch.com/Default.aspx), final PEG concentration was 8.5% for trimer preparations and 12% for monomer preparations. The green precipitate was resuspended in 30 mM Tricine-NaOH, pH 8, 0.05% DM, (10 mM NaCl for monomer; 75 mM NaCl for trimer), and loaded onto a 10–30% sucrose density gradient, prepared in 30 mM Tricine-NaOH pH-8, 0.05% DDM (10 mM NaCl for monomer; 75 mM NaCl for trimer). Following centrifugation (SW40 rotor, 37,000 rpm, 12 hr) the appropriate green band was collected and used for kinetic measurements or crystallization.

## Photosystem I crystallization

The chlorophyll peak from the sucrose gradient was collected, NaCl was added to a final concentration of 150 mM and the complex was precipitated using PEG3350 (9% for trimeric complexes; 12% for monomeric complexes). After centrifugation (13,000 rpm, 5 min in an Eppendorf tabletop), the green precipitate was resuspended in minimal volume of buffer (3 mM Tricine-NaOH pH 8, 0.02% DDM), and any undissolved material was removed by repeating the centrifugation step. The chlorophyll concentration in the soluble material was adjusted to the desired concentration (typically 3 mg/ml) using the above buffer, and dispensed in 4 µl drops into 24-well, sitting drop crystallization plates. Protein drops were mixed 1:1 with reservoir solution (90 mM NaCl, 90 mM MgCl$_2$, 0.0005% n-*Nonyl*-β-D-*Maltoside* (NM), 100 mM glycine, 30 mM Tricine-NaOH, pH 8, 9–11% gradient of PEG3350 for trimer complex; 60 mM Tricine-NaOH, pH 8, 9% PEG3350, 0.0005% NM, 100 mM glycine, 100–120 mM NaCl gradient for monomer complex). Crystals were formed within 1–3 days, and grown to their final size by a week. The crystals of the PSI trimer were large and flat with no precipitate in the well. In contrast, the monomeric complex formed multiple crystal forms. Only a small fraction of the crystals, recognized through their overall shape, diffracted to high resolution. The other crystals diffracted to very low resolution or did not diffract at all. Cryo protection was achieved by replacing the mother liquor in the well and in the reservoir with solutions containing progressively increasing PEG3350 concentrations (10%, 15%, 20% and 30%) with a minimum of 2 hr between each solution (usually 24 hr between each step). Finally crystals were placed in 40% PEG3350 solution for a few minutes, mounted on loops and frozen in liquid nitrogen.

## Data collection and refinement

Data were collected at ESRF, Grenoble, station ID29, or at the SLS, Villigen, station PXI. Diffraction frames were processed with XDS (*Kabsch, 2010*), scaled and merged with SCALA (*Evans, 2006*; *Winn et al., 2011*).

## Monomer

Two data sets from two different crystals were used in the refinement of the monomeric complex, one obtained from ID29 at the ESRF and one from PXI at SLS. Refinement was done with PHENIX (*Adams et al., 2010*). Monomer descriptions of chlorophyll and β-carotene were computed from average values of the 1.9 E PSII structure (*Umena et al., 2011*). At 3 Å the use of additional, external restraints was found to be ineffective. TLS refinement was used throughout the refinement. A very coarse choice of TLS groups was found to be optimal with the membrane domain, the stromal ridge and PsaK chosen as three separate TLS groups. Adding hydrogen atoms to the model was found to greatly improve the clash score and geometry of the model and so hydrogens were used throughout the refinement.

## Trimer

The PSI[PsaJF] model was refined against a single data set obtained at ID29. Initially a truncated model (with most side chains and phytol tails removed) of PSI from *Thermosynechococcus* was used for molecular replacement and as a source of external restraints. Refinement was carried out using Refmac5 (*Murshudov et al., 2011*), using a coarse TLS model including only two groups, one encompassing the entire membrane domain and the other defined as the stromal ridge (PsaC, PsaD and PsaE). The use of external restraints (as well as NCS restraints) was vital to achieving reasonable statistics and ProSMART (*Nicholls et al., 2012*) was found to be the most effective choice in our case. Multiple rounds of refinements followed by manual building by coot were carried out. PHENIX was used in the final rounds solely for B factor and TLS refinement. The use of the higher resolution *Synechocystis* monomeric model resulted in a ~1.5% reduction in R values (both working and free) and more importantly resulted in better maps. The improvement in map quality allowed us to complete the tracing of PsaK (without side chains), which is very disordered in our monomeric model and was not traced completely even in the high-resolution structure from *Thermosynechococcus*.

## Cytochrome c553 purification

The soluble fraction after membrane sedimentation was used as a starting material for purification. Soluble material from 100 l culture was first clarified using 50% ammonium sulfate precipitation, than the remaining material in the supernatant was precipitated using 100% ammonium sulfate. The precipitated material was dialyzed over-night and loaded on a DEAE column equilibrated with 10 mM Tris–HCl pH 7.4. The column was eluted using a linear NaCl gradient (0–200 mM), pink fractions were collected and concentrated using a centricon (5 kDa cutoff). The concentrated pink fraction was loaded on a Superdex 75 gel filtration column. The pink protein peak coming off this column contained a single 10 kDa polypeptide according to SDS-PAGE and displayed the appropriate absorbance spectrum for cytochrome c553.

## P700 reduction kinetics

Kinetic measurements were carried out using a JTS-10 spectrophotometer. Reaction mixture (for the data presented in *Figure 3A,B*) included 35 µg chl of purified PSI, 60 mM MES pH-6, 0.02 %DDM, 25 mM $MgCl_2$, 50 mM NaCl, 5 mM ascorbate, 5 µM methyl viologen and cytochrome C553 (1 µM) or horse heart cytochrome c (2.5 µM) in a total volume of 1 ml. A 100 ms light pulse was applied from an orange LED source (630 nm) and three independent traces were averaged for each experiment.

## Light-dependent cytochrome oxidation

Absorbance changes (550 nm–540 nm) were recorded using an Aminco DW-2 spectrophotometer. The components were mixed in a 1 ml quartz cuvette and they included 40 µg chl of purified PSI, 30 mM Tricine-HCl pH 8, 25 mM NaCl, 10 mM ascorbate, 10 µM methyl viologen, 5 mM horse heart cytochrome C or 2.5 mM cytochrome C553. Light was provided from a slide projector and filtered through Appropriate filters.

## Acknowledgements

Yuval Mazor would like to thank Ofer Rog for critically reading the manuscript. The authors would like to thank the ESRF, SLS and BESSYII synchrotrons for beam time and the staff scientists for excellent guide and assistance.

## Additional information

### Funding

| Funder | Grant reference number | Author |
| --- | --- | --- |
| European Research Council | 293579-HOPSEP | Yuval Mazor, Daniel Nataf, Hila Toporik, Nathan Nelson |
| Israel Science Foundation | 204-10 | Yuval Mazor, Daniel Nataf, Hila Toporik, Nathan Nelson |

The funders had no role in study design, data collection and interpretation, or the decision to submit the work for publication.

### Author contributions

YM, NN, Conception and design, Acquisition of data, Analysis and interpretation of data, Drafting or revising the article; DN, HT, Acquisition of data, Analysis and interpretation of data

## Additional files

### Supplementary files

• Supplementary file 1. List of primers used in the construction of *PsaJF*. *PsaJ* was amplified with primers 5580 and 5581 and fused to *PsaF* fragment amplified with primers 5582 and 5583. The *JF* fusion gene was than ampified using ~300 bp sequences containing the *PsaF* promoter and the *PsaF* down homology to create the entire gene cassette. This 970 bp fragment was cloned into pGEM-Teasy. Finally, the fragment was moved into a pET28 vector in order to introduce the *Kan* resistance gene into a PstI site that was designed into primer 5584.

### Major datasets

The following datasets were generated:

| Author(s) | Year | Dataset title | Dataset ID and/or URL | Database, license, and accessibility information |
|---|---|---|---|---|
| Mazor Y, Nataf D, Toporik H, Nelson N | 2013 | Crystal structure of a virus like photosystem I from the cyanobacterium *Synechocystis* PCC 6803 | 4L6V; http://www.rcsb.org/pdb/search/structidSearch.do?structureId=4L6V | Publicly available at the RCSB Protein Data Bank (http://www.rcsb.org/). |
| Mazor Y, Nataf D, Toporik H, Nelson N | 2013 | Crystal structure of a virus like photosystem I from the cyanobacterium *Synechocystis* PCC 6803 | 4KT0; www.rcsb.org/pdb/search/structidSearch.do?structureId=4KT0 | Publicly available at the RCSB Protein Data Bank (http://www.rcsb.org/). |

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
