## [Decision Letter]

Thank you for sending your work entitled “Crystal structures of virus-like photosystem I complexes from the mesophilic cyanobacterium *Synechocystis* PCC 6803” for consideration at *eLife*. Your article has been favorably evaluated by a Senior editor and 3 reviewers, one of whom, Werner Kühlbrandt, is a member of our Board of Reviewing Editors, and one of whom, James Barber, has agreed to reveal his identity.

The Reviewing editor and the other reviewers discussed their comments before we reached this decision, and the Reviewing editor has assembled the following comments to help you prepare a revised submission.

This is an excellent and well-written paper. Through X-ray crystallography it explores two intriguing facets of photosystem I. (i) It explores the possible role played by the prevalence of photosynthetic PSI genes in marine cyano phages genomes and (ii) it seeks to understand the structural differences between PSI of mesophilic (*Synechocystis* sp. PCC 6803) and thermophylic cyanobacteria (*Thermosynechococcus elongatus*) in relation to their optical and functional properties.

With regard to the first question they have constructed, using genetic engineering, a phage-like PSI (removal of the N terminus of PsaF and fusion psaJ and psaF genes) and determined its structure at 3.8Å. Apparently the consequential and significant conformational changes that occur on the electron donor results in a promiscuous PSI complex able to utilize electrons from a wider range of soluble cytochromes that are normally functional only in respiration.

The second question has been explored by purifying and crystallizing the monomeric form of PSI from *Synechocystis* sp. PCC 6803. Here the resolution of the structure is better at 2.8Å. Differences between this structure and that published for PSI isolated from *Thermosynechococcus elongatus* are discussed with particular focus on the identification of long wavelength absorbing chlorophylls known as “red chlorophylls”. Several important differences have been identified to explain why *Thermosynechococcus* has a higher content of red chlorophylls than *Synechocystis*.

Fascinating details include the possible identification of the notorious “red” chlorophylls in PS-I, a conserved lipid involved in chlorophyll coordination, and the conserved water cluster in the electron transfer path.

Nevertheless, the manuscript needs to be revised to address the following concerns:

1) The presence of entire PS-I genomes in marine cyanophages is intriguing, but the discussion of its biological significance, and of the structure of the phage-mimetic PS-I complex, in this manuscript is slightly disappointing. The manuscript would benefit from a more in-depth discussion of the possible role of the virally-encoded PSI in present-day organisms and in PS-I evolution, rather than referring obliquely to things that are “not at all clear” or “a matter of debate”. What is unclear and what is the debate? What would it take to clarify the issues and resolve the debate?

2) Figure 1 is presumably an SDS gel. If so, please state this in the figure legend. Why then does PsaF run at almost twice the mass of the PsaJF fusion protein? Should the fusion protein not be substantially larger than PsaF? Where is PsaJ on the wt gel?

3) Is Figure 5 depth-cued? If yes, please say so in the figure legend. If not, what are the shadowy chlorine rings? Assuming that it is depth-cued, there are at least 4 chlorophylls that are present in the *Thermosynechococcus* but not in the *Synechocystis* structure. How can we be sure that these are not the red chlorophylls of *Thermosynechococcus*?

4) The functional or structural role of the conserved water clusters in the electron transfer path deserves a more detailed discussion.

5) An outline of how the viral PSI was constructed in *Synechocystis* should be provided. How were the PsaJ and PsaF genes replaced by the single, viral PsaJF gene? Did the authors remove the original PsaJ and PsaF genes, and then insert the PsaJF gene? If this is the case, where was the PsaJF gene inserted? Did the authors also remove the PsaL and PsbI genes, as they were not present in the viral operon?

---

## [Author Response]

*1) The presence of entire PS-I genomes in marine cyanophages is intriguing, but the discussion of its biological significance, and of the structure of the phage-mimetic PS-I complex, in this manuscript is slightly disappointing. The manuscript would benefit from a more in-depth discussion of the possible role of the virally-encoded PSI in present-day organisms and in PS-I evolution, rather than referring obliquely to things that are “not at all clear” or “a matter of debate”. What is unclear and what is the debate? What would it take to clarify the issues and resolve the debate*?

We have made significant expansions in our discussion on these points, and tried to be as clear and as explicit as possible.

*2)*
Figure 1
*is presumably an SDS gel. If so, please state this in the figure legend. Why then does PsaF run at almost twice the mass of the PsaJF fusion protein? Should the fusion protein not be substantially larger than PsaF? Where is PsaJ on the wt gel*?

We have modified the legend to clearly indicate the use of a SDS gel in the figure. The JF fusion is actually ∼20 amino acids shorter than PsaF. In addition, the soluble domain of PsaF was exchanged with the single TM helix of PsaJ and the final protein is much more hydrophobic than PsaF alone. This is the reason for the unusual migration of this polypeptide. We have verified by sequencing and MS analysis that both the DNA and the expressed protein conform to our design and of course the presence of a correct length fusion protein in the structure provides the final proof in this matter. PsaJ as well as the other small subunits of PSI are not resolved in this particular gel setup.

*3) Is*
Figure 5
*depth-cued? If yes, please say so in the figure legend. If not, what are the shadowy chlorine rings? Assuming that it is depth-cued, there are at least 4 chlorophylls that are present in the Thermosynechococcus but not in the Synechocystis structure. How can we be sure that these are not the red chlorophylls of Thermosynechococcus*?

Figure 5 is indeed depth cued and we have now indicated this fact in the legend. We are not sure to which chlorophyll molecules this comment refers to. There are three chlorophylls coordinated by PsaL which are missing from the monomeric structure, and three other chlorophylls are missing from the PSI^PsaJF^ model. We have made a composite image that includes all chlorophylls from the high resolution structure and added to it the PsaL coordinated chlorophylls to give a complete picture of the antenna arrangement. In some cases the superposition is close enough so that the rings occupy essentially the same position, which can be confusing. A single peripheral chlorophyll (in addition to B33) is absent from *Synechocystis* (numbered M1601 in *Thermosynechococcus,* in the upper left corner of Figure 5). Strictly speaking this chlorophyll can be red chlorophyll as well. However, our discussion of red chlorophylls is based on chlorophylls pairs, which, based on their ring to ring distances, may be electronically coupled. In that sense chlorophyll 1601 is distant and cannot be affected by other pigments. Other mechanisms for controlling the excitation energy of chlorophylls, such as their interaction with amino acid side chains, cannot be currently predicted and require experimental verification that is outside the scope of the current work.

*4) The functional or structural role of the conserved water clusters in the electron transfer path deserves a more detailed discussion*.

We have expanded to some extent our discussion on the water cluster. However determination of its potential importance requires additional experiments and computational methods that are well beyond the scope of the current work.

*5) An outline of how the viral PSI was constructed in Synechocystis should be provided. How were the PsaJ and PsaF genes replaced by the single, viral PsaJF gene? Did the authors remove the original PsaJ and PsaF genes, and then insert the PsaJF gene? If this is the case, where was the PsaJF gene inserted? Did the authors also remove the PsaL and PsbI genes, as they were not present in the viral operon*?

We included a more detailed description of the construction of the PsaJF strain. In short, since PsaF and PsaJ are adjacent to each other in the genome of *Synechocystis* we replaced both genes with the PsaJF gene under the control of the native PsaF promoter. The phrasing of the comment suggests that we used the viral PsaJF gene and we would like to clarify that we fused the *Synechocystis* PsaJ and PsaF genes. Other labs conveyed to us that they were unsuccessful in generating a functional PsaJF strain using the viral gene. We used *Synechocystis* genes in order to avoid additional problems in the integration of the modified subunit into the PSI complex and address the question of removing the PsaF N terminus in a more precise manner. A similar strategy was employed, for example, when the importance of the positive patch in the PsaF N terminus was addressed using a chimeric gene made of the N terminus of *Chlamydomonas* PsaF fused to the PsaF gene of *Synechococcus*.

PsaL and PsaI are not deleted in the strain used to prepare the monomeric PSI. However, both polypeptides are absent from the complex at very early stages of our preparation (Figure 4—figure supplement 1) and again the final model leaves no doubt as to their absence. This was a fortuitous discovery made during our attempts to crystalize the complete monomeric form of PSI that are ongoing.